# Dynamic Characteristic Monitoring of Wind Turbine Structure Using Smartphone and Optical Flow Method

**Wenhai Zhao** [1] , **Wanrun Li** [1,2,3,*] , **Boyuan Fan** [1] **and Yongfeng Du** [1,2,3]

1 Institution of Earthquake Protection and Disaster Mitigation, Lanzhou University of Technology, Lanzhou 733050, China
2 International Research Base on Seismic Mitigation and Isolation of GANSU Province, Lanzhou University of Technology, Lanzhou 733050, China
3 Disaster Prevention and Mitigation Engineering Research Center of Western Civil Engineering, Lanzhou University of Technology, Lanzhou 733050, China
* Correspondence: ce_wrli@lut.edu.cn

**Abstract:** The dynamic characteristics of existing wind turbine structures are usually monitored using contact sensors, which is not only expensive but also time-consuming and laborious to install. Recently, computer vision technology has developed rapidly, and monitoring methods based on cameras and UAVs (unmanned aerial vehicles) have been widely used. However, the high cost of UAVs and cameras make it difficult to widely use them. To address this problem, a target-free dynamic characteristic monitoring method for wind turbine structures using portable smartphone and optical flow method is proposed by combining optical flow method with robust corner feature extraction in ROI (region of interest). Firstly, the ROI region clipping technology is introduced after the structural vibration video shooting, and the threshold value is set in the ROI to obtain robust corner features. The sub-pixel displacement monitoring is realized by combining the optical flow method. Secondly, through three common smartphone shooting state to monitor the structural displacement, the method of high pass filtering combined with adaptive scaling factor is used to effectively eliminate the displacement drift caused by the two shooting states of standing and slightly walking, which can meet the requirements of structural dynamic characteristics monitoring. After that, the structural displacement is monitored by assembling the telephoto lens on the smartphone. The accuracy of displacement monitored by assembling the telephoto lens on the smartphone is investigated. Finally, the proposed monitoring method is verified by the shaking table test of the wind turbine structure. The results show that the optical flow method, combined with smartphones, can accurately identify the dynamic characteristics of the wind turbine structure, and the smartphone equipped with a telephoto lens is more conducive to achieving low-cost wind turbine structure dynamic characteristics monitoring. This research can provide a reference for evaluating the condition of wind turbine structures.

**Keywords:** structural health monitoring; wind turbine structure; smartphone; optical flow method

## 1. Introduction

As wind energy stands out among many renewable energy sources, people are paying more and more attention to the safety performance of wind turbine structures [1,2]. The amount of wind energy obtained is related to wind turbine blades. In order to obtain more wind energy and generate more power, the size of wind turbine blades has grown exponentially in recent years [3,4]. Wind turbine blades may be defective or damaged due to production defects, turbulent winds, lightning, irregular loads, etc. [5], which may lead to surface changes that affect the aerodynamic efficiency of the blades [6,7], thus causing serious safety problems. As the main component of wind turbine to capture wind energy, the power generation efficiency and safety of wind turbine structure mainly depends on the health status of the blades. To ensure that the wind turbine structure will not be damaged,



thus effectively reducing economic losses, it is of great significance to monitor the vibration response of the wind turbine structure.

The traditional wind turbine detection is mainly manual, which is not only costly and unsafe, but also requires the experience of engineers, so it is not suitable for efficient detection of wind turbine structures. In addition, ultrasonic, thermal imaging, telescope, and other equipment are often used for wind turbine structure monitoring [8,9]. However, due to its high cost and the high professional level required for the use of instruments, it has not been widely used in the detection of wind turbine structures. Damage detection based on vibration analysis is mainly carried out by changing dynamic characteristic parameters [10,11]. Therefore, it is meaningful to identify dynamic characteristic parameters of wind turbine structure through vibration data. Civil engineering monitoring mainly uses contact sensors to monitor dynamic characteristics [12]. However, the disadvantage of using sensors is that they are fixed to the surface of the structure. This not only causes damage to the structure, but also substantially modifies the structure mode due to the mass load effect [13]. These wind turbines are huge structures, so conventional methods cannot be used for monitoring. Some wind turbines operate in complex natural environments, including wind, salt fog, rain, etc. Thus, the sensor-based wind turbine structure monitoring faces different types of challenges, such as sensor damage, extensive wiring, and labor intensity. Data acquisition is still challenging despite the large number of non-contact sensors emerging in structural health monitoring. Therefore, monitoring of the structural dynamic characteristics has not seen its popularity [14].

Computer vision-based methods for structural health monitoring have been proposed and applied in practice by many scholars [15–18]. Engineers favor computer vision technology with application advantages, such as non-contact, long distance, fast, low cost, and low labor to routine operations. Feng et al. [19] performed a vision-based measurement of dense full-field displacement with simply supported beams using the template-matching algorithm. It was verified practically when the trains were passing though the bridge. Dong et al. [20] used feature point extraction and optical flow tracking algorithm to identify dynamic characteristics of the stadium stand compared with contact sensors. This overcame the problem of small sensor monitoring area to achieve multi-point measurement. Khadka et al. [21] used digital image correlation (DIC) method to perform target tracking on wind turbine blades and used marker points to identify dynamic characteristics of wind turbine blades. Song et al. [22] perfectly segmented the background through the depth learning full convolution network (FCN) and conditional random field (CRF), and then used DIC to measure the displacement. The feasibility of this method was verified by experiments under different illumination conditions. Through the research of a large number of scholars, computer vision is feasible as a supplement to traditional structural health monitoring.

A large number of structural dynamic characteristics identification methods based on computer vision use cameras or unmanned aerial vehicles. However, because of the high cost of cameras or UAVs, low-cost equipment is needed to monitor the structural dynamic characteristics through the same imaging difference with the camera. In recent years, smartphones have developed on an unprecedented scale and can now be used as an effective measurement tool in structural health monitoring [23]. Most researchers use the internal accelerometers of smartphones in combination with the actual structure for structural health monitoring [24–26]. However, such methods require tying bind smartphones to structures and the smartphone can only monitor data from one point, so they are greatly limited in structural dynamic characteristics. Zhao et al. [27] developed an APP: D-viewer that can monitor bridge displacements by using color-matching algorithm and smartphones, and conducted static and dynamic tests on the bridge. However, the article does not consider the complex backgrounds, and the monitoring is not effective under lighting conditions, which is not conducive to long-term monitoring. Zhao et al. [28] proposed a new visual cable force measurement method based on smartphone cameras, and preliminarily verified its feasibility and practicability through cable model tests. Li et al. [29] provided the possibility of crowd perception of all buildings in urban areas after the earthquake

with smartphone-based monitoring technology. Ozer et al. [30] introduced the concept of smartphone structural health monitoring and constructed a hybrid structure vibration response measurement framework by using multi-sensor smartphone functions. A novel hybrid motion sensing platform has been successfully implemented through the integration of various sensor types and devices. Wang et al. [31] developed an automatic damage detection system based on smartphones which can realize real-time damage detection of masonry buildings through experimental verification. The research on the monitoring of structural dynamic characteristics by cheap and commonly used smartphones was initially minimal. It is still limited to the monitoring of artificial markers and single environments at present. Therefore, it is necessary to explore a smartphone-based method for monitoring the dynamic characteristics of wind turbine structures which is suitable for low-cost, target-free structural dynamic characteristics monitoring.

This paper focuses on the research of monitoring the dynamic characteristics of wind turbine blades based on the combination of computer vision and smartphones. In Section 2, a smartphone-based target-free wind turbine structure dynamic characteristics monitoring method was proposed using the optical flow method and ROI clipping. After shooting the structural vibration videos, ROI was selected to set the threshold value to obtain robust corner features and combined with the optical flow method to achieve sub-pixel displacement monitoring. In Section 3, camera calibration and vibration tests were conducted for different models of smartphones to verify the feasibility of smartphone monitoring. Through the analysis of three common smartphone shooting states to monitor the structural displacement, the method of high pass filtering combined with adaptive scaling factor was used to effectively eliminate the displacement drift caused by the two shooting states of standing and slightly walking to monitor the structural displacement. After that, the structural displacement was monitored by assembling the telephoto lens on the smartphone, and the accuracy of displacement monitored by assembling the telephoto lens on the smartphone was investigated. In Section 4, the proposed monitoring method was verified by using a small shaking table test. The results show that computer vision combined with smartphones can accurately identify the natural frequency of the wind turbine structure, and that a smartphone equipped with a telephoto lens is more conducive to achieving low-cost wind turbine structure dynamic characteristics monitoring. Finally, the modal shape of wind turbine structure is obtained. Section 5 summarizes the research content of this paper.

## 2. Dynamic Displacement Monitoring Based on Computer Vision

Computer vision detects, extracts, recognizes, and tracks moving objects in image sequences to obtain moving object parameters. The dynamic characteristics monitoring of wind turbine structures using smartphones and visual algorithms consists of four parts: camera calibration, feature recognition, target tracking, and jitter processing and displacement calculation.

### 2.1. Camera Calibration

In recent years, the lenses of modern consumer cameras have been significantly improved. However, cameras tend to be smaller and more convenient. Generally, these cameras use wide-angle lenses. Smartphone cameras increase the field of vision by introducing significant radial distortion. In order to eliminate this distortion and obtain accurate displacement measurements with the consumer camera, the camera must be corrected.

Generally, the calibration process is divided into two steps. The first step is to convert the world coordinate system to the camera coordinate system. This step is to convert the three-dimensional points to three-dimensional points, including the relevant parameters of external camera parameters such as $R$, $t$. The second step is to convert the camera coordinate system to the image coordinate system. This step is to convert the 3D points to 2D points, including the relevant parameters of the $K$ isoperimetric camera internal parameters. The camera calibration steps are shown in Figure 1. The synchronous calibration of internal

and external parameters generally includes two strategies: optical calibration, that is, using known geometric information (such as fixed length checkerboard) to achieve parameter solution. Another strategy is called self-calibration, that is, using the structural motion in static scenes to estimate the parameters.

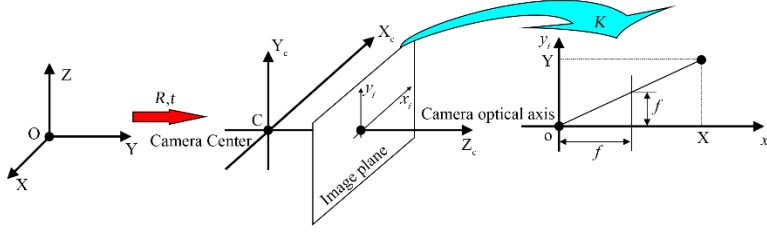

**Figure 1.** General steps for camera calibration.

Camera calibration is to take the calibration plate by the camera, determining the internal and external parameters of the camera with the intrinsic value of the calibration plate's feature points. Following that, the image coordinates are converted to physical coordinates through the scale factor. The following equation is used to convert image coordinates to physical coordinates:

$$\begin{pmatrix} x \\ y \\ 1 \end{pmatrix} = \begin{bmatrix} f_x & \gamma & c_x \\ 0 & f_y & c_y \\ 0 & 0 & 1 \end{bmatrix} \begin{bmatrix} r_{11} & r_{12} & r_{13} & t_1 \\ r_{21} & r_{22} & r_{23} & t_2 \\ r_{31} & r_{32} & r_{33} & t_3 \end{bmatrix} \begin{bmatrix} X \\ Y \\ Z \\ 1 \end{bmatrix} \tag{1}$$

The simplified expression is:

$$sx = K[R|t]X \tag{2}$$

where $s$ is the scale factor; $(x, y, z, 1)^{\mathrm{T}}$ is the image coordinate; $K$ is the camera internal parameter; $(X, Y, Z, 1)^{\mathrm{T}}$ is the world coordinates; $f_x$ and $f_y$ are the focal lengths of the camera in the horizontal and vertical directions; $c_x$ and $c_y$ are the offsets of the optical axis; $\gamma$ is the tilt factor; $R$ and $t$ are the camera the external parameters; $r_{ij}$ and $t_i$ are elements of $R$ and $t$, respectively.

The camera's internal parameters, tangential distortion, and radial distortion are used to calibrate the video, which can effectively eliminate lens distortion and image distortion, thus obtaining more accurate displacement measurements. In this paper, the smartphone lens uses Zhang's calibration method to calibrate the camera lens, followed by carrying out the video calibration [32].

*2.2. Target Tracking Principle Based on Optical Flow Method*

Optical flow is an assumption based on the image brightness motion information. The optical flow calculation is based on two assumptions about the optical characteristics of object motion: constant brightness assumption and small motion assumption. Assume that a pixel $(x, y)$ on the image has a corresponding brightness of $I(x, y, t)$ at time $t$, and a brightness of $I(x + dx, y + dy, t + dt)$ at time $t$. From the optical flow consistency assumption:

$$I(x, y, t) = I(x + dx, y + dy, t + dt) \tag{3}$$

The basic equation of optical flow can be obtained by using Taylor expansion Equation (3):

$$\frac{\partial I(x, y, t)}{\partial x} u + \frac{\partial I(x, y, t)}{\partial y} v + \frac{\partial I(x, y, t)}{\partial t} = 0 \tag{4}$$

where $u = dx/dt$ and $v = dy/dt$ are the instantaneous velocities of pixels in the image in $x$ and $y$ directions at time $t$.

Assuming $I_x = \partial(x, y, t)/\partial x$, $I_y = \partial(x, y, t)/\partial y$ and $I_t = \partial(x, y, t)/\partial t$, it can be converted into the optical flow constraint equation:

$$I_x u + I_y v + I_t = 0 \tag{5}$$

Since one of the above optical flow constraint equations cannot solve the two unknowns $(x, y)$, it is necessary to establish a new constraint equation to solve it. In 1981, the Lucas Kanade (LK) optical flow method [33], proposed by Lucas and Kanade, put forward the assumption of spatial consistency to solve the basic equation of optical flow.

The LK optical flow method assumes that the motion vector of the neighborhood $\Omega$ in a space is constant. In a neighborhood $\Omega$ of $n$ pixels, each pixel satisfies the following equation:

$$I_{xi} u + I_{yi} v + I_{ti} = 0 \quad i = 1, 2, \cdots, n \tag{6}$$

At this time, the constraint equation of optical flow can be changed into:

$$E_c(u, v) = \iint (I_x u + I_y v + I_t)^2 dx dy \tag{7}$$

In neighborhood $\Omega$, the error of LK optical flow is:

$$E_{LK}(u, v) = \iint W^2(x, y) \cdot (I_x u + I_y v + I_t)^2 dx dy \tag{8}$$

where $W(x, y) = \{w_i | i = 1, 2, \cdots n\}$ is the weight value of each point in the field, and the farther away from the center point, the smaller the corresponding weight value.

Discretization of Equation (8) leads to:

$$\begin{bmatrix} \sum\limits_i^n w_i^2 I_{xi}^2 & \sum\limits_i^n w_i^2 I_{xi} I_{yi} \\ \sum\limits_i^n w_i^2 I_{xi} I_{yi} & \sum\limits_i^n w_i^2 I_{yi}^2 \end{bmatrix} \begin{bmatrix} u \\ v \end{bmatrix} = - \begin{bmatrix} \sum\limits_i^n w_i^2 I_{xi} I_{ti} \\ \sum\limits_i^n w_i^2 I_{yi} I_{ti} \end{bmatrix} \tag{9}$$

where $I_{xi}$, $I_{yi}$, and $I_{ti}$ are the gradient values corresponding to the $x$, $y$, and $z$ directions of the first pixel in the neighborhood, respectively; $w_i$ is the weight of the $i$th pixel. Assumptions:

$$A = \begin{bmatrix} I_{x1}, \cdots, I_{xn} \\ I_{y1}, \cdots, I_{yn} \end{bmatrix}^T \tag{10}$$

$$W = diag(w_{x1}, w_{x2}, \cdots, w_{xn}) \tag{11}$$

$$b = [I_{t1}, I_{t2}, \cdots, I_{tn}]^T \tag{12}$$

Equation (7) can be expressed as:

$$A^T W^2 A \begin{bmatrix} u \\ v \end{bmatrix} = A^T W^2 b \tag{13}$$

Then the vector optical flow can be calculated:

$$\begin{bmatrix} u \\ v \end{bmatrix} = (A^T W^2 A)^{-1} A^T W^2 b \tag{14}$$

Finally, the corresponding pixel position $(x, y)$ of each target point is calculated from the vector optical flow $u = dx/dt$ and $v = dy/dt$.

### 2.3. Target-Free Feature Extraction Based on Optical Flow Method

The optical flow method can track the target in a low-resolution image. When the vibration is small, the error is small. Moreover, the optical flow method occupies less

computer memory to achieve fast calculation. Therefore, the optical flow method can be used to monitor the displacement response of wind turbine structures. Although the optical flow method can effectively track the structural feature points, the optical flow method requires that the feature points must be prominent to be recognized effectively. In the actual monitoring of wind turbine structure, it is impossible to place artificial markers on the structures. To solve this problem, this paper proposes a robust corner feature extraction method based on region of interest (ROI) to realize dynamic characteristics monitoring of target-free structures.

In the optical flow method, the detector is usually used to extract the feature point as Harris corner [34], which has been widely used in engineering practice [35,36]. Since Harris corners use a Gaussian filtering, and the motion speed is relatively slow, there is a risk of corner information loss and information migration. Therefore, tracking errors will occur in optical flow tracking. J. Shi improved Harris corner algorithm in 1994 and proposed an improved Shi–Thomasi corner detection operator [37]. The Shi–Tomasi corner operator solves the problem of feature point aggregation by limiting the shortest distance between two interest points, and only allows points beyond a certain distance to become interest points from the strongest point of Harris corner. It solves the defect of Harris corner and is more suitable for optical flow tracking. Based on this, this paper conducts ROI selection before structural vibration video processing, and then sets the quality factor and threshold value for the number of Shi–Thomasi corners. Pause at the first frame when the video starts playing, and the operator manually selects the ROI. The selected ROI should not only include the range of structural vibration, but also avoid excessive image selection to avoid excessive calculation. Finally, corners with better robustness appear on the structure in the ROI area for target tracking, and sub-pixel coordinate extraction is achieved by capturing image corners and combining them with the optical flow equation. The basic flow of target tracking based on ROI target-free robust corner feature extraction is shown in Figure 2.

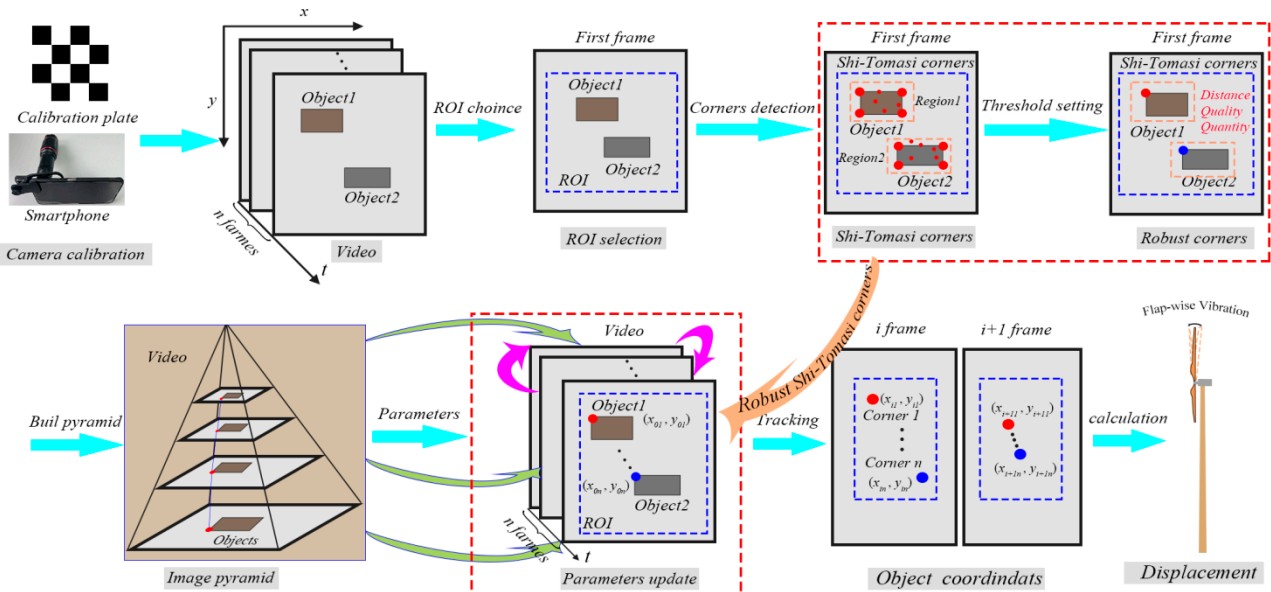

**Figure 2.** Target tracking based on ROI for target-free robust corner feature extraction.

Step 1: Establish the region of interest (ROI). After taking the vibration videos, the smartphone selects the ROI in the first frame, which aims to reduce corner recognition and improve the recognition accuracy of the required corner area.

Step 2: Shi–Tomasi corner detection. Based on the first frame of the video, all Shi–Tomasi corners within the ROI are detected.

Step 3: Select the corner. Not all the corners detected in step 2 are required due to engineering requirements. Select the corner with the strongest feature in the corresponding

area or multiple corners containing the structure itself for target tracking through such constraints as corner distance, number of corners, quality factor, etc.

Step 4: Build an image pyramid. In view of the small motion assumption among the three assumptions of the optical flow method, the image is sampled down in the form of an upper pyramid to compensate for excessive motion. In this paper, considering the actual working conditions and smartphone lens frame rate, a 4-layer pyramid was built for each frame image.

Step 5: Update the parameters. After each image pyramid is built to meet the optical flow conditions, the strongest corner selected in the first frame will be used for feedback to the next frame. In each frame of image, the corners to be tracked are the corners of the previous frame, so it is necessary to detect the corners of each frame, and then compare them with the corners of the first frame to update the corner parameters.

Step 6: Target tracking. After updating the parameters, optical flow correlation calculation can be carried out according to the corners to obtain the corresponding corner coordinates of each frame. The coordinates extracted from the optical flow equation are sub-pixel coordinates.

In the target tracking based on optical flow method, except for the first step where the ROI area needs to be manually selected, the other steps are automatically carried out, laying a foundation for displacement monitoring in structural dynamic characteristics monitoring.

### 2.4. Smartphone Jitter Processing

Although smartphones can be fixed with tripods for visual monitoring, considering the convenience of using smartphones, measures should be taken counteract the effects of common smartphone shooting methods on the monitoring of structural dynamic characteristics. When the hand-held smartphone is used for shooting, it will shake, which is similar to the situation when the UAV is monitoring. Li et al. [38] used the method of in-plane high pass filtering and out-of-plane adaptive scaling factor to deal with the jitter of UAV hovering monitoring when using UAV to monitor the wind turbine blades. This paper uses similar methods to deal with the jitter of smartphones.

The high pass filter filters the low frequency noise by passing the measurement signal across the high pass filter and can restore the projection of the displacement on the still image plane to scaling and perspective factors. Therefore, it is expressed in the matrix form as follows:

$$\begin{bmatrix} x_{i_{hp}} \\ y_{i_{hp}} \end{bmatrix} = \begin{bmatrix} s_x & p_{yx} \\ p_{xy} & s_y \end{bmatrix} \begin{bmatrix} X_{world} \\ Y_{world} \end{bmatrix} \tag{15}$$

where $x_{i_{hp}}$ and $y_{i_{hp}}$ respectively represent the results of high pass measurement signals.

The out-of-plane smartphone jitter processing adopts the method of adaptive scaling factor, and its equation is as follows:

$$S_i = \frac{L}{l} = \frac{L}{\|p_{i1} - p_{i2}\|} \tag{16}$$

where $S_i$ is the scale factor at the time of the $i$th frame, $L$ is the actual distance between two structural points, and $p_{i1}$ and $p_{i2}$ are the image coordinates of $p_1$ and $p_2$ on the image at the time of the $i$th frame.

### 2.5. Displacement Calculation

The corresponding coordinate $P_i(x_i, y_i)$ of the target in each frame of the time series and the coordinate $P_0(x_0, y_0)$ of the target in the first frame are obtained by the optical flow method of target tracking. Following that, the absolute displacement of the structure was obtained by converting the image coordinate to the physical coordinate through the adaptive scale factor. Considering the elevation problem when using smartphones to

capture videos, the displacement $d_S$ of the structure was calculated by angle correction using the following equation:

$$d_S = S_i \times \frac{P_i - P_0}{\cos^2 \theta} \tag{17}$$

where $\theta$ is the angle between the smartphone optical axis and the measured target.

Through the above target tracking and displacement calculation, the absolute displacement of the structure can be calculated. Following that, the displacement responses of multiple monitoring points of the structure are obtained through monitoring. Finally, the modal shape of the wind turbine structure can be calculated by responses.

## 3. Smartphone Performance Test

### 3.1. Smartphone Lens Distortion Test

This paper uses iPhone 12 and Honor X10 smartphones to investigate the performance of the proposed method. The camera parameters of the two phones are shown in Table 1.

**Table 1.** Parameters of two smartphone cameras.

| Smartphone Category | Smartphone Photo | Frame Rate | Max Pixel | Pixel Density | Zoom Multiple | Aperture |
|---|---|---|---|---|---|---|
| iPhone 12 | | 30/60 fps | 12 million | 460 ppi | 5 | f/2.4 |
| Honor X10 | | 30/60 fps | 40 million | 397 ppi | 10 | f/1.8 |

Due to the process error of the camera optical system during manufacturing, there is geometric distortion between the actual imaging and the ideal imaging. Distortion is mainly divided into radial distortion and tangential distortion, as shown in Figure 3.

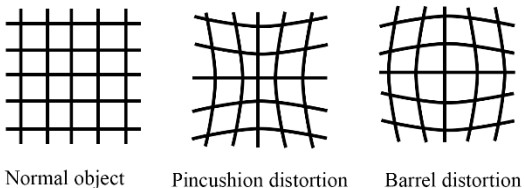

Normal object     Pincushion distortion     Barrel distortion

**Figure 3.** Normal image and distorted image.

The radial distortions (pillow and barrel) are mainly caused by the lens quality and the fact that the light is more bent away from the center of the lens than near the center. Generally, the radial distortion can be corrected by the following equation:

$$x_{corr} = x_{dis}(1 + k_1 r^2 + k_2 r^4 + k_3 r^6) \tag{18}$$

$$y_{corr} = y_{dis}(1 + k_1 r^2 + k_2 r^4 + k_3 r^6) \tag{19}$$

Tangential distortion (thin lens distortion and centrifugal distortion) is caused by defects in lens manufacturing that make the lens itself not parallel to the image plane. Tangential distortion can be corrected by the following equation:

$$x_{corr} = x_{dis} + \left[ 2p_1 xy + p_2(r^2 + 2x^2) \right] \tag{20}$$

$$y_{corr} = y_{dis} + \left[ p_1(r^2 + 2y^2) + 2p_2 xy \right] \tag{21}$$

where $x_{corr}$ and $y_{corr}$ represent the coordinates of the image plane in $x$ and $y$ directions after repair, respectively, $x_{dis}$ and $y_{dis}$ represent the coordinates of the image plane in $x$ and $y$

directions with distortion, respectively; $k_1$, $k_2$, and $k_3$ are radial distortion parameters; $p_1$ and $p_2$ represent tangential distortion parameters.

It can be seen that the image distortion has $k_1$, $k_2$, $k_3$, $p_1$, and $p_2$, totaling five parameters. For a camera with good quality, the tangential distortion is small, and the radial distortion coefficient can be ignored. Only two parameters need to be calculated. For a camera with good quality, the tangential distortion is very small and can be ignored. The radial distortion coefficient $k_3$ can also be ignored. Only $k_1$ and $k_2$, need to be calculated.

Zhang's calibration method can be carried out in the integrated package of MATLAB. Theoretically, only two calibration photos are needed to calibrate the camera parameters. In order to verify the accuracy of the camera parameters identified by the calibration photos taken by smartphones, 33 marking photos taken by a smartphone were used for calibration. The calibration plate was a 25 mm × 25 mm black and white checkerboard. The process of using two smartphones to identify radial distortion parameters is shown in Figure 4.

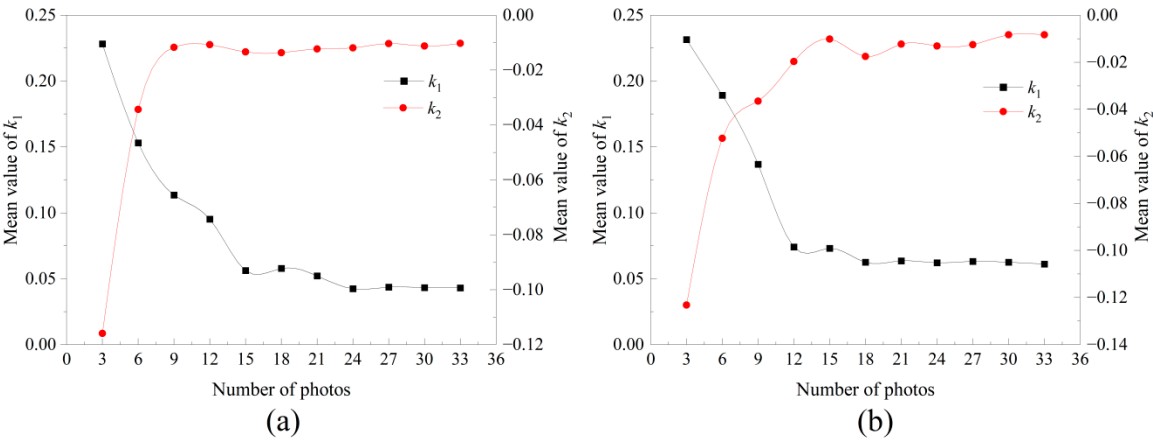

**Figure 4.** Recognition trend of radial distortion coefficient: (**a**) Parameter recognition of Honor X10 smartphone; (**b**) iPhone 12 smartphone parameter recognition.

It can be seen from Figure 4 that the accuracy of parameter calibration tends to be stable with the increase of the number of calibrations during the process of camera calibration parameter identification for the lens of smartphone using the calibration plate. The parameter calibration accuracy of smartphones is worse than that of cameras, but the calibration requirements can be met by using about 20 photos. Finally, through the camera calibration program, the radial distortion parameters $k_1$ and $k_2$ of the Honor X10 smartphone are 0.0431 and −0.0102, respectively. The radial distortion parameters $k_1$ and $k_2$ of the iPhone 12 smartphone are 0.0611 and −0.0835, respectively.

The smartphone lens was adjusted to a wide angle to form the distortion, and then distortion parameters were used to correct the image, as shown in Figure 5. The distorted and distorted edges of the black and white squares of the image completed by distortion parameter correction have been well-corrected to straight edges. Therefore, the camera calibration method can be used to correct the image distortion in smartphone image acquisition to obtain more accurate monitoring.

### 3.2. Smartphone Displacement Monitoring

In the case of simulated shutdown, smartphones were used for vibration monitoring of scaled wind turbine models, and an LDS (laser displacement sensor) was used to evaluate the accuracy of visual data. Canon R6 camera and 24–105 mm zoom lens were also used in the test. The test apparatus is shown in Figure 6.

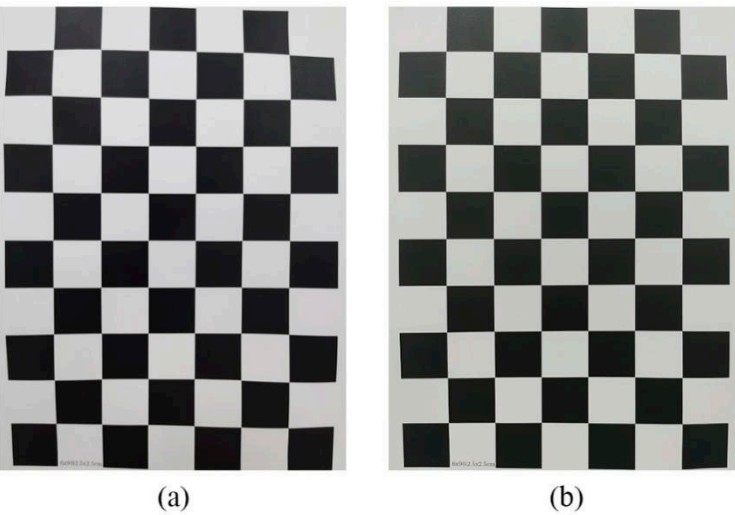

**Figure 5.** Smartphone image correction: (**a**) Before correction; (**b**) After correction.

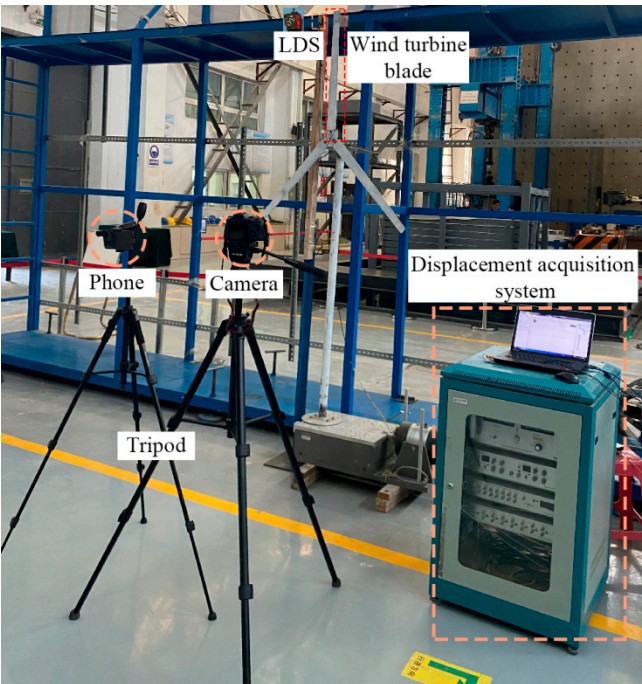

**Figure 6.** Smartphone displacement monitoring test equipment.

In practical wind turbine structure monitoring state, the background is relatively complex, and the structural health monitoring usually using the optical flow method will be affected by the background [39]. In order to verify that the ROI-based corner feature extraction method of unmarked robustness realizes the dynamic characteristics monitoring of unmarked structures, the test has verified the feature-matching between simple backgrounds and complex backgrounds, as shown in Figure 7.

It can be seen from Figure 7 that, no matter in a simple background or in a complex background, corner-matching using the feature-matching method after selecting ROI is very effective, and there is no error-matching. Therefore, it is feasible to use optical flow method to select ROI for target-free dynamic characteristic monitoring. In the actual monitoring process, multi-point monitoring is realized by selecting multiple ROIs for monitoring at the same time, which solves the limitation that one sensor can only monitor one point in traditional monitoring.

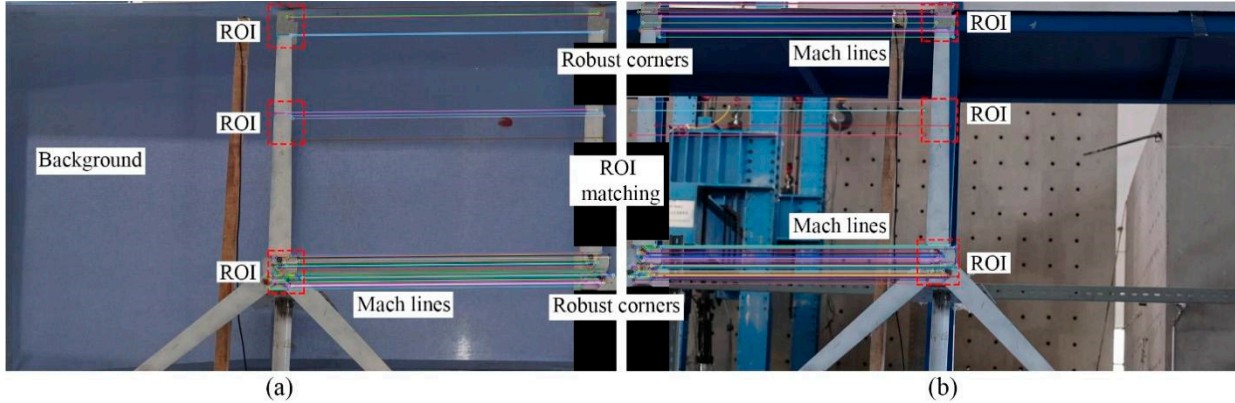

**Figure 7.** ROI based target-free feature-matching: (**a**) Simple background; (**b**) Complex background.

To verify the accuracy of the smartphone to monitor the displacement, the wind turbine blade was released to free vibration after given initial displacement. The LDS is used as the data reference, and the camera is used for verification. The smartphone adopts 1080 × 1920 resolution with a frame rate of 60 fps. The frequency of LDS is set to 50 Hz. The resolution of 1080 × 1920 is adopted with a frame rate of 50 fps. The camera and mobile phone are fixed with a tripod. Through 100 s of data acquisition, the displacement time history is shown in Figure 8.

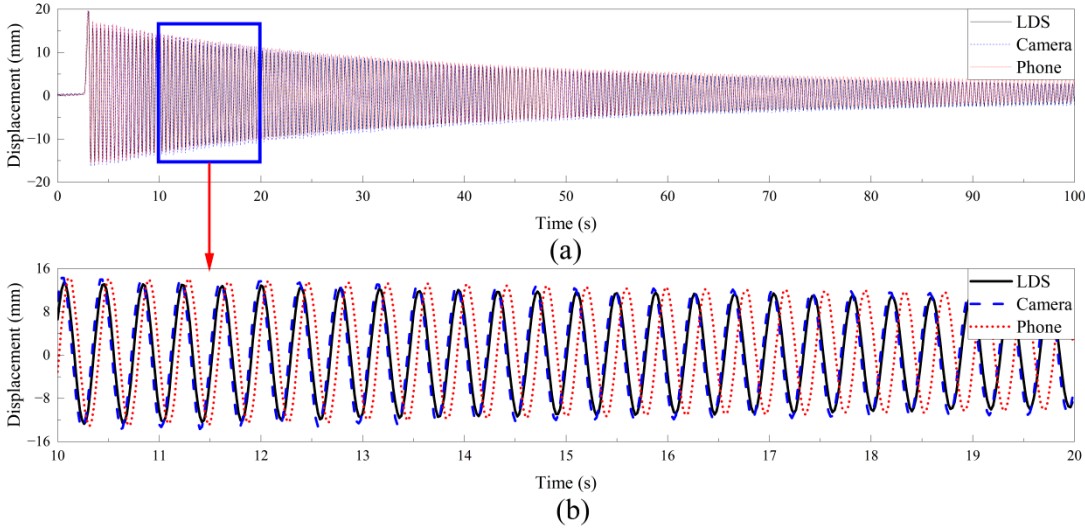

**Figure 8.** Time history comparison of vibration displacement monitored by smartphones: (**a**) Whole displacement time history; (**b**) 10~20 s displacement time history.

It can be seen from Figure 8a that the displacement time history through vision is generally consistent with that of LDS. Figure 8b shows that the camera and LDS are basically consistent in the 10~20 s displacement time history details. However, the phase is inconsistent smartphone monitoring during operation. Therefore, such problems must be addressed to enable smartphones to accurately monitor displacement. The frame rate is usually used for time conversion in visual monitoring. However, due to the instability of the sensor when the smartphone captures video, the captured video will not be consistent with the original set frame rate. For example, if 60 fps is set, the final captured video frame rate is 59.58 fps or 60.56 fps. This is also the reason why the phase of the visual displacement curve and the standard displacement curve is not consistent in displacement monitoring. To solve this problem, the video output is read at the real-time frame rate by traversing all frames, and finally time conversion is performed at the real-time read frame rate. The displacement time history curve after frame rate correction is shown in Figure 9.

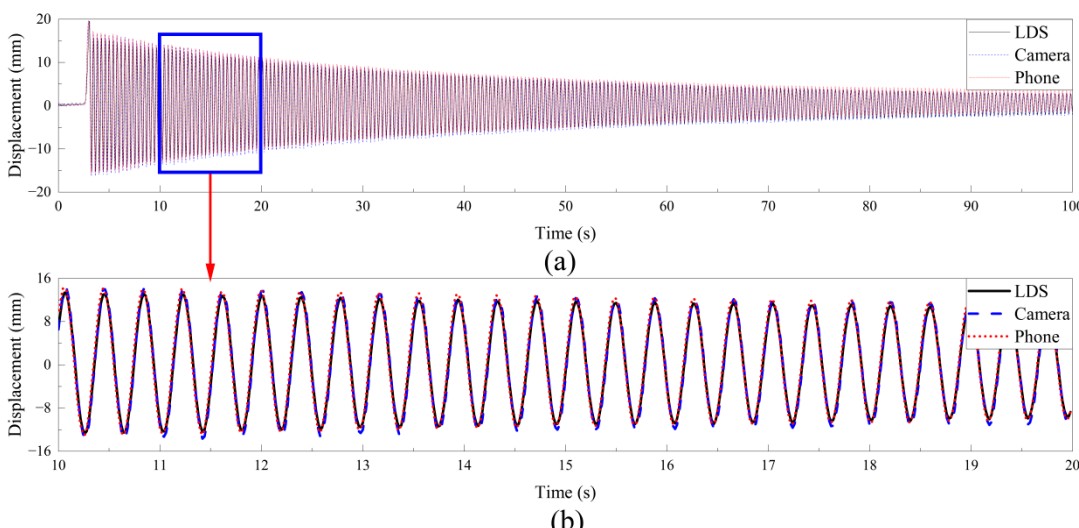

**Figure 9.** Time history comparison of smartphone displacement monitoring through frame rate correction: (**a**) Whole displacement time history; (**b**) 10~20 s displacement time history.

It can be seen from the time domain information shown in Figure 9 that the problem of phase difference during smartphone monitoring can be effectively solved by frame rate correction, which is consistent with the displacement curve monitored by the camera. Through the free vibration test of wind turbine blades, it was proved that the method of selecting target-free displacement monitoring by smartphones combined with ROI is effective.

### 3.3. Performance Test of Smartphones in Different States

As one of the indispensable tools in human life, smartphone monitoring has the advantages of high efficiency and low cost. However, people usually do not carry tripods when they travel, so if they want to use smartphones more conveniently to complete structural monitoring tasks, they must explore monitoring methods for convenient photography. When using a smartphone to take pictures of objects without a tripod, people tend to stand still or walk slightly. Therefore, to monitor the dynamic characteristics of the structure based on smartphones without a tripod, the test uses three states to explore: smartphones on the tripod, holding smartphones when standing still, and using smartphones when walking slightly. The test used the same smartphone (Honor X10) to shoot fixed points in three shooting states: tripod, standing handheld smartphone, and slightly walking handheld smartphone. The monitoring statue under the three shooting states was inversely deduced from this stationary fixed point. In order to better distinguish the three shooting states, the direction coordinates centered on the smartphone are specified in this paper. The schematic diagram of the tester's shooting and the smartphone direction are shown in Figure 10.

The tester stood at the fixed point 2 m away and photographed the fixed point for 4 min. The displacement time histories in three directions under three shooting conditions are shown in Figure 11, and the maximum displacement is shown in Table 2.

It can be seen from Figure 11 that only the tripod is stable in three directions under three conditions, and the last section also proves the feasibility of displacement monitoring with support. Standing hand-held photography and light walking photography have large displacement in three directions—the displacement deviation of light walking in Z-direction is especially large. In the test, the moving distance is controlled within 0.5 m when shooting with slightly walking, so the displacement deviation of this kind of shooting will be larger in actual situations. Table 2 shows the three shooting states using smartphones and the displacement peaks in three directions, from which it can be seen that slightly walking has the largest displacement in three directions. The test data processing adopts the

pixel as the unit, and the farther the distance is, the greater the actual error is in the actual monitoring. Therefore, if the smartphone is used to monitor the dynamic characteristics of the structure, the displacement offset under the common states must be processed.

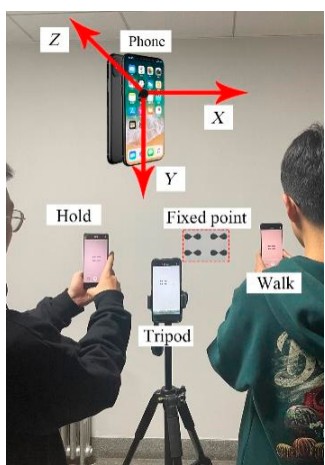

**Figure 10.** Directions for testers to take pictures and smartphones.

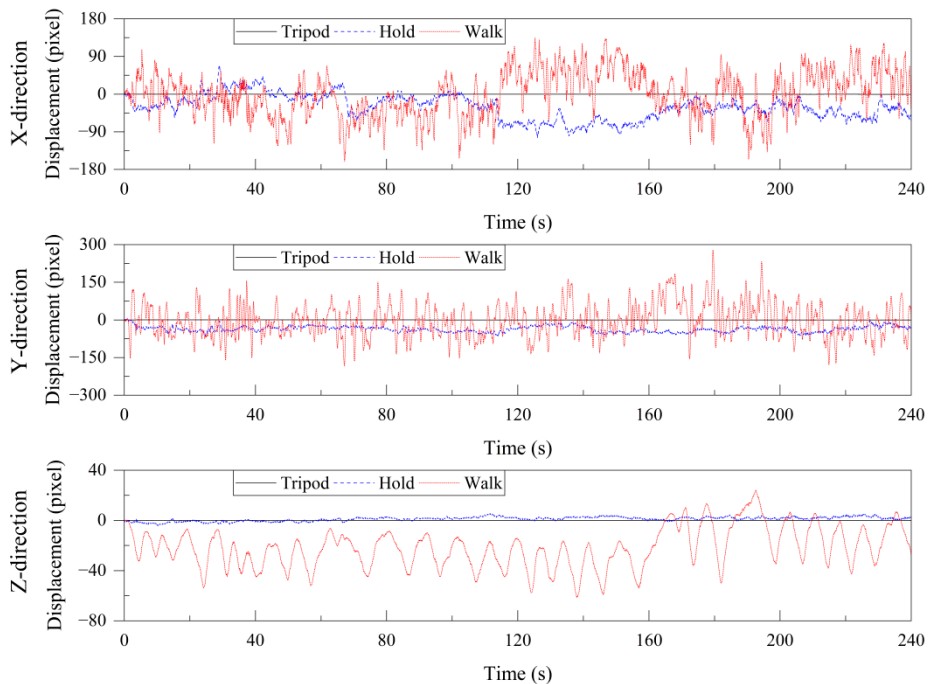

**Figure 11.** Displacement time histories comparison in three directions of three shooting states using a smartphone.

**Table 2.** Displacement peak value under different conditions when using a smartphone.

| Motion State | X-Direction/Pixel | Y-Direction/Pixel | Z-Direction/Pixel |
|---|---|---|---|
| Equipped with tripod | 0.015 | 0.034 | 0.018 |
| Standing shooting | 95.025 | 72.183 | 6.254 |
| Walk slightly | 150.641 | 282.944 | 61.239 |

The frequency domain information obtained by processing the displacement time history data captured in Figure 11 through Fast Fourier Transform (FFT) is shown in

Figure 12. The X-direction data are used in the tripod equipped and hand-held shooting states, and the Z-direction data are used in the slightly moving shooting state.

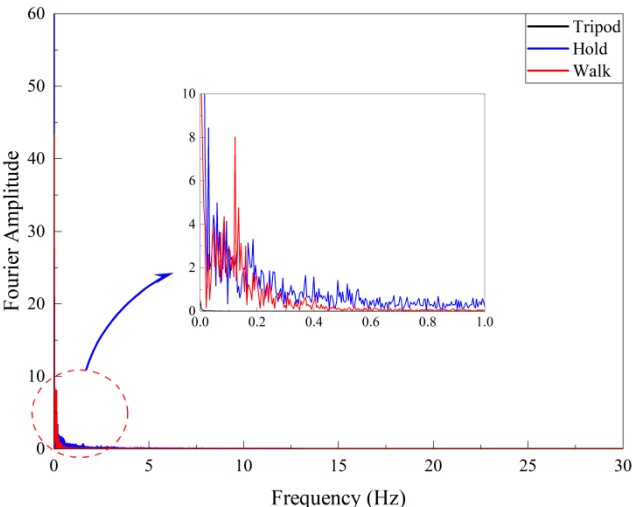

**Figure 12.** Frequency domain comparison of three shooting states using smartphone.

It can be seen from the frequency domain information taken in the three states shown in Figure 12 that the frequency domain of the shooting method with a bracket is basically 0, and the frequency domain information of the shooting state of standing, holding, and walking slightly is confused before 0.6 Hz. Therefore, high pass filtering can be used to eliminate the displacement error caused by different shooting states of smartphones.

### 3.4. Structural Displacement Monitoring Using Smartphone in Different States

Since the above tests have verified the reliability of smartphone and LDS displacement monitoring, the test will hammer the blade to make it vibrate freely, and use three shooting states: tripod, standing hand-held smartphone, and slightly walking hand-held smartphone to monitor the displacement.

Taking the displacement time history obtained by the shooting method equipped with a tripod as a benchmark, the displacement time history monitored by the two shooting states of standing hold smartphone and slightly walking hand-held smartphone is shown in Figures 13 and 14.

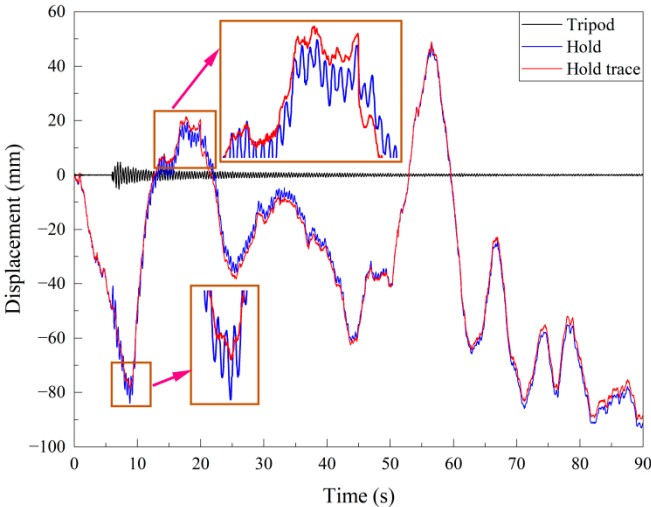

**Figure 13.** Displacement time history comparison in the state of standing hold smartphone shooting.

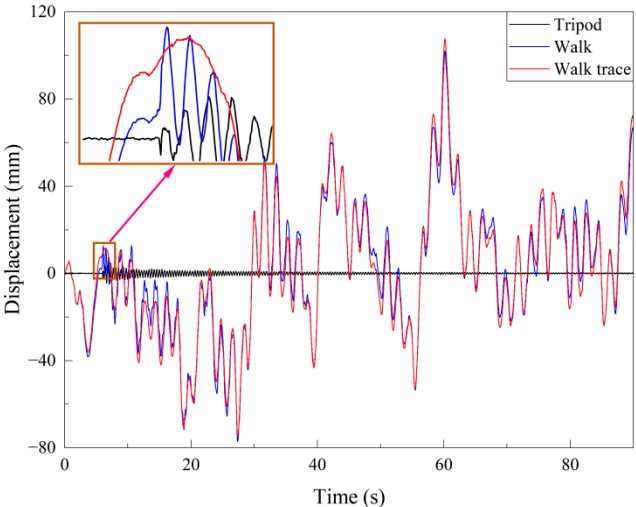

**Figure 14.** Displacement time history comparison under the shooting state of slightly walking hand-held smartphone.

Figures 13 and 14 show the displacement time history comparison between the standing hold and slightly walking shooting and the tripod equipped shooting states. Although the displacement time histories of the two states are scattered, the two trajectories inversely calculated from the fixed points can match the structural vibration displacement time histories. When the trajectories shown in Figures 13 and 14 are consistent with the displacement drift path, high pass filtering can be used for noise processing. In this paper, a high pass filter with a cut-off frequency of 0.6 Hz was used to denoise smartphones. Combined with the adaptive scaling factor method, the final processing results are shown in Figure 15.

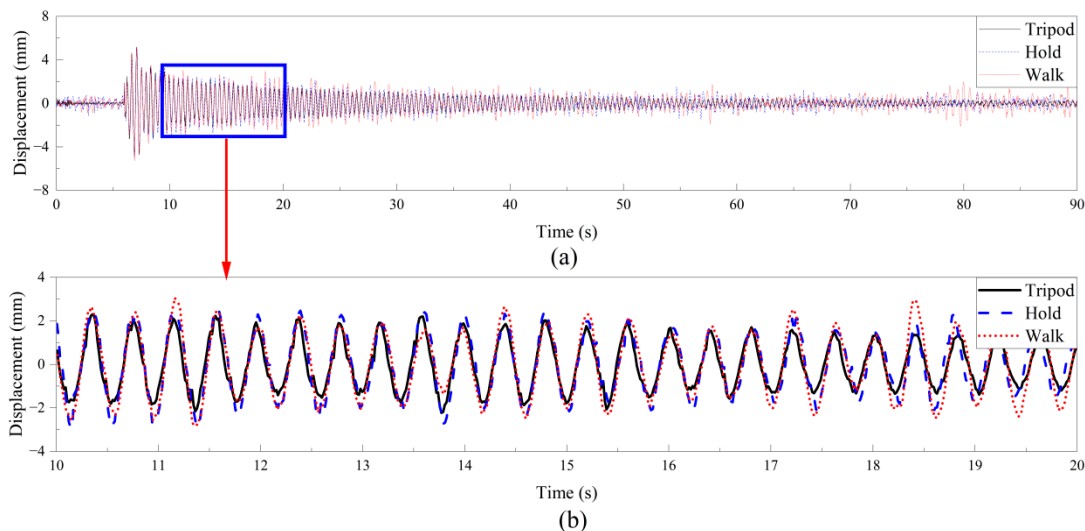

**Figure 15.** Comparison of position shift time history de-noising using smartphone in different states: (**a**) Whole displacement time history; (**b**) 10~20 s displacement time history.

It can be seen from Figure 15a that the displacement time history of the monitoring method with support can be basically consistent with that of the method using high pass filtering combined with adaptive scaling factor. The influence of displacement is not completely eliminated in the area with small displacement. Since the displacement is relatively small, it often presents low-frequency movement, and the displacement time history is consistent on the whole, which does not affect the actual monitoring. The 10 ~ 20 s displacement time history details show that the displacement is eliminated better by the above methods when the handheld smartphone is shooting, and the amplitude is still

inconsistent in some areas when the walking camera is shooting. The displacement drift in the walking process was not completely filtered out due to the influence of the tester's walking during the processing.

To quantify the errors of the two common ways of holding smartphones to monitor the dynamic characteristics of structures, this paper uses Root Mean Square Error (RMSE), correlation coefficient ($\rho$), and determination coefficient ($R^2$) for error analysis. The equations are:

$$\text{RMSE} = \sqrt{\sum_i \left(x_v(i) - x_s(i)\right)^2 / n} \tag{22}$$

$$\rho = \frac{\left|\sum_i \left(x_s(i) - \mu_s\right) \times \left(x_v(i) - \mu_v\right)\right|}{\sqrt{\sum_i \left(x_s(i) - \mu_s\right)^2}\sqrt{\sum_i \left(x_v(i) - \mu_v\right)^2}} \tag{23}$$

$$R^2 = 1 - \frac{\sum_i \left(x_v(i) - x_s(i)\right)^2}{\sum_i \left(x_s(i) - \mu_s\right)} \tag{24}$$

RMSE is calculated using Equation (22), where $n$ is the total number of monitoring, and $x_v$ and $x_s$ are displacement data from vision monitoring and laser displacement sensors, respectively. $\rho$ calculated using Equation (23), where $\mu_v$ and $\mu_s$ are the average values of the two displacement trajectories. The calculation equation of $R^2$ is Equation (24), which is used to determine the matching degree of the two recorded tracks [40]. The error comparison of vibration monitoring data of two common smartphone shooting methods is shown in Table 3, and the error distribution is shown in Figure 16, in which the displacement time history data with tripod is taken as the benchmark.

**Table 3.** Displacement monitoring error of common smartphone shooting methods.

| Shooting Method | RMSE | $\rho$ | $R^2$ |
|---|---|---|---|
| Standing shooting | 0.6219 | 0.8254 | 0.8763 |
| Walk slightly | 0.7342 | 0.7513 | 0.7925 |

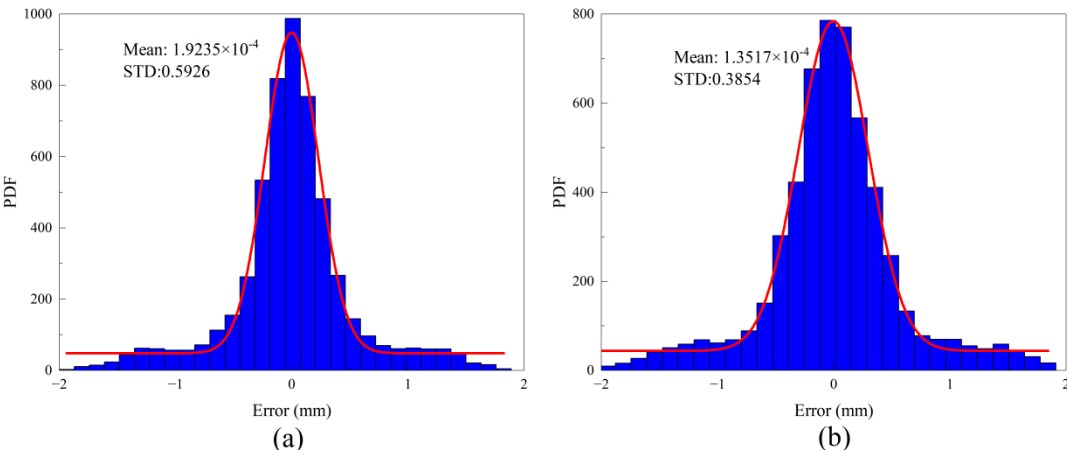

**Figure 16.** The error distribution of displacement monitoring under the common smartphone shooting mode: (**a**) Standing shoot; (**b**) Shoot when walking slightly.

It can be seen from Table 3 that the errors of standing shoot and slightly walking shooting are within the acceptable range. The maximum error shown in Figure 16 does not exceed 2 mm, and most of the errors are concentrated within 0.5 mm, which is acceptable in time domain monitoring. The displacement time history data monitored by the three shooting methods are transformed by FFT, as shown in Figure 17.

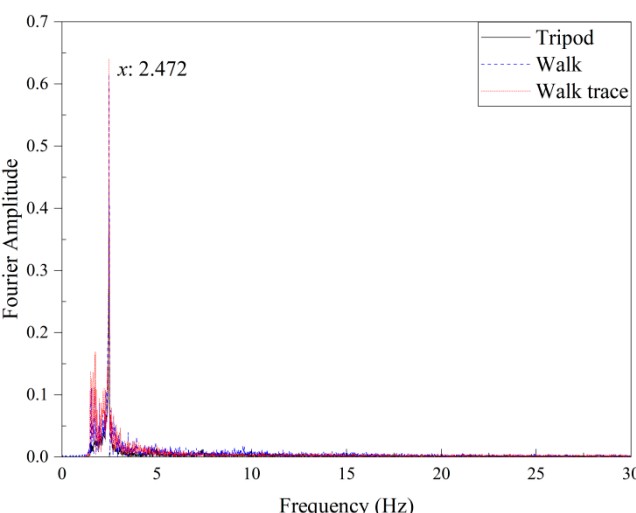

**Figure 17.** Frequency domain comparison of monitoring under three common smartphone shooting states.

It can be seen from the comparison of monitoring frequency domains under the three common shooting states shown in Figure 17 that, although there is a certain error between the hold smartphone and the slightly walking shooting state in the time domain, the natural frequency identified in the frequency domain is consistent. Therefore, it is feasible to monitor the displacement of structures by three common shooting methods in the frequency domain. Through the method of high pass filtering and adaptive scaling factor, the structural dynamic characteristics can be accurately monitored by removing the noise from displacement monitoring of hand-held smartphones and shooting methods of slightly walking.

### 3.5. Structural Displacement Monitoring of Smartphone Assembled with Long Focus Lens

The actual wind turbine structure is enormous, and the use of ordinary smartphones cannot obtain clear images due to lens limitations, so accurate monitoring cannot be achieved. The lens is an important part of the camera, but with the development of technology and the improvement of technology, lenses are not limited to cameras. Many manufacturers produce telephoto lenses that can be matched on smartphones. By assembling a telephoto lens in a smartphone, the defect that the smartphone cannot take pictures of distant objects can be effectively solved. Figure 18 shows the contrast of the experimenter using a camera and a smartphone equipped with a telephoto lens to shoot the wind turbine at a distance of 5 m from the wind turbine. The shooting area of the camera in Figure 18a is wider than that of the telephoto lens equipped with the smartphone in Figure 18b, but the longer objects can be shot through the smartphone. Figure 18b is just a picture taken in a smartphone with one pixel and one distance of the telephoto lens. The telephoto lens can be up to 32 times as long as possible and can shoot objects thousands of meters away. The connection between the smartphone and the telephoto lens can be easily fixed by a clamp. Therefore, the displacement of the actual wind turbine structure can be monitored by assembling a telephoto lens on a smartphone.

Through the hammer test on the wind turbine, the smartphone was equipped with a telephoto lens to monitor the blade tip. Limited by the test site, the smartphone was 10 m away from the wind turbine, and the monitoring structure is shown in Figure 19.

It can be seen from Figure 19a that the displacement monitored by the telephoto lens is consistent with that monitored by the camera. The displacement time history monitored by the smartphone equipped with the telephoto lens is consistent with the LDS at both the phase and peak. Therefore, smartphones can be used to assemble telephoto lenses for structural dynamic characteristics monitoring, which enriches the way smartphones monitor dynamic characteristics.

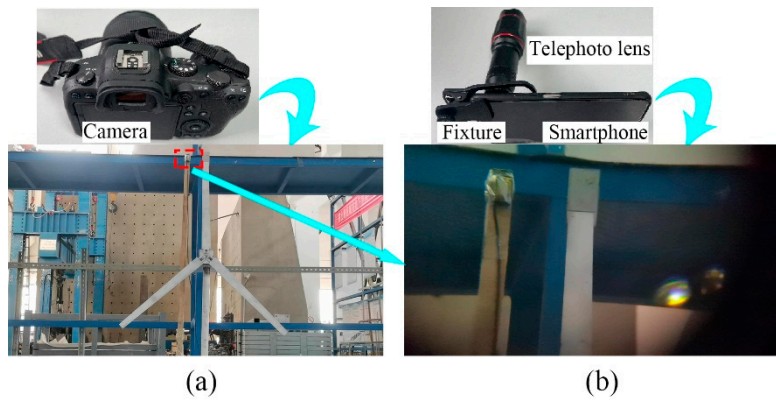

**Figure 18.** Comparison between camera shooting and smartphone equipped telephoto lens shooting: (**a**) Camera; (**b**) Smartphone equipped with telephoto lens.

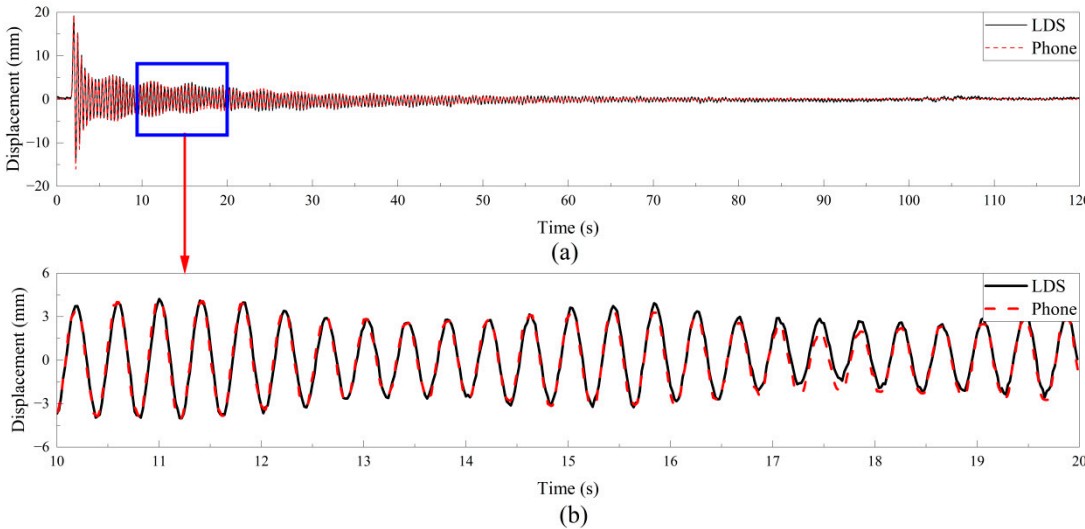

**Figure 19.** Comparison of displacement time history of telephoto lens for smartphone assembly: (**a**) Whole displacement time history; (**b**) 10~20 s displacement time history.

## 4. Dynamic Characteristic Monitoring of Wind Turbine Structure

### 4.1. Experimental Equipment

To monitor the dynamic characteristics of the wind turbine structure, the test uses the Honor X10 smartphone, with a lens resolution of up to 4K, a maximum frame rate of 60 fps, a video resolution of 1080P, and a 32 times telephoto lens. At the same time, in order to verify the reliability of smartphones, the Canon R6 camera was also used in the test. The frame rate is 50 fps, and a 24 mm~105 mm zoom lens was used for test verification. Zhang's camera calibration method was used for video shooting to correct lens distortion. The vibration test was conducted on the scaled wind turbine model under simulated shutdown. The arrangement of test device and measuring points is shown in Figure 20. In order to verify the accuracy of the visual test, 1 LDS and 5 accelerometers were used to monitor the structural responses, where the sampling frequency is set to 50 Hz and the traditional vibration test digital system is used for verification.

### 4.2. Natural Frequency Identification of Wind Turbine Structure

The vibration of the wind turbine structure has two directions: edgewise and flap-wise. Since the flap-wise direction vibration belongs to out-of-plane vibration, the flap-wise direction is more likely to cause damage to the wind turbine structure. Monitoring the dynamic characteristics of wind turbine structure flap-wise direction is the basis for ensuring the normal operation of wind turbine structure.

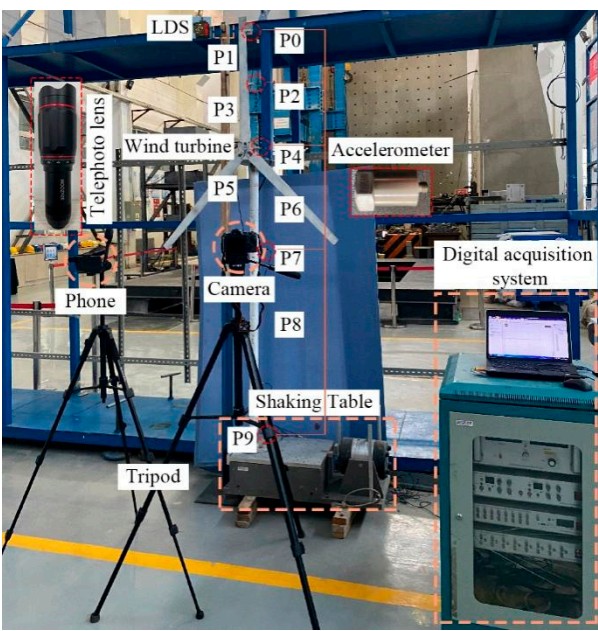

**Figure 20.** Experimental equipment.

The blade tip of the scaled wind turbine model is hammered in the flap-wise direction to make it vibrate freely. The acceleration data during the vibration were monitored by the accelerometers, and the camera and LDS were used as the displacement reference for verification. The acceleration time history curve monitored by the accelerometer is shown in Figure 21. The domain information, monitored by smartphones, cameras, and LDS, is shown in Figure 22.

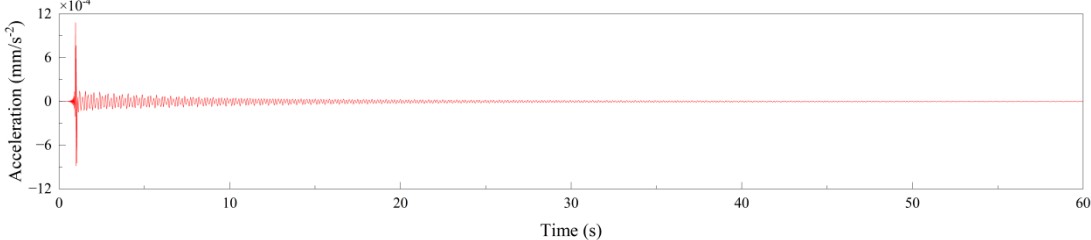

**Figure 21.** Acceleration time history in flap-wise direction.

It can be seen from Figures 21 and 22 that the acceleration time history monitored by the accelerometer after hammering the blade tip attenuates rapidly, while the displacement monitored by LDS and vision attenuates slowly. The main reason is that the accelerometer is attached to the structure and the monitoring frequency is high, so the data monitored by the acceleration sensor tends to zero after the blade vibration slow down. The displacement time histories show, in Figure 22, that the displacement time histories monitored by camera and smartphone are consistent with LDS as a benchmark. Therefore, the use of smartphones can achieve low-cost dynamic characteristics monitoring of wind turbine structures. The displacement time history monitored by accelerometer, LDS, and smartphone is converted into PSD frequency domain information as shown in Figure 23. In order to make the images more intuitive, the horizontal axis of the coordinates is expressed in logarithmic coordinates.

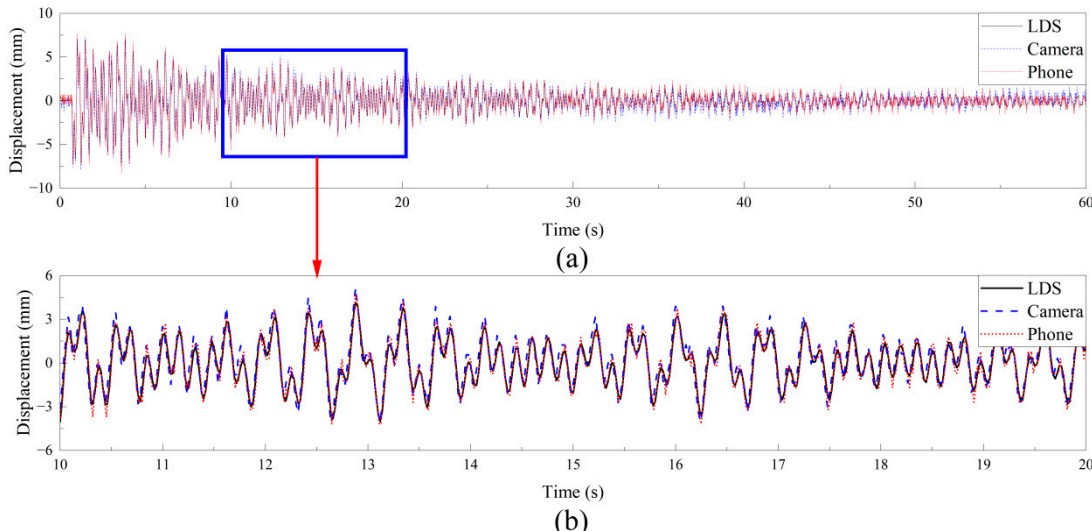

**Figure 22.** Time history comparison of vibration displacement in flap-wise direction: (**a**) Whole displacement time history; (**b**) 10~20 s displacement time history.

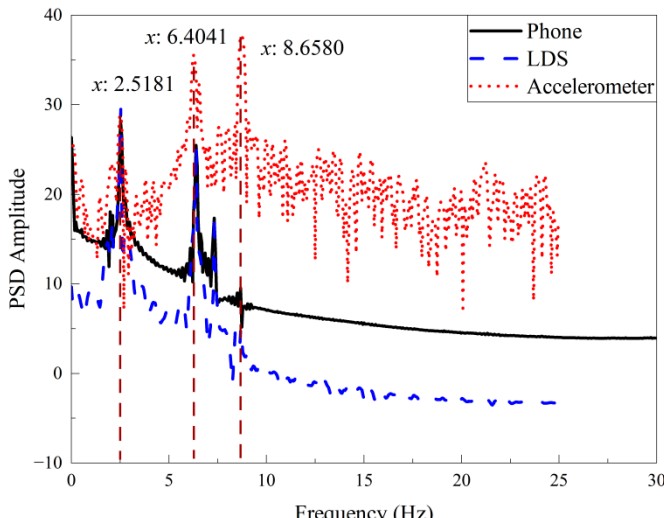

**Figure 23.** PSD comparison of different equipment.

Figure 23 shows that the natural frequencies obtained by monitoring the wind turbine structure with three types of equipment are consistent. The first three natural frequencies of the wind turbine structure under simulated shutdown in the flap-wise direction are 2.5181 Hz, 6.4041 Hz, and 8.6580 Hz, respectively. In Figure 23, the amplitude of the first three natural frequencies recognized by the accelerometer increases sequentially, while the third natural frequencies recognized by LDS and smartphones are fuzzy. Through frequency domain comparison, the monitoring data of structural dynamic characteristics represented by smartphones are consistent with LDS. The smartphone can identify the natural frequency of the structure stably through its camera, especially for low order frequencies. The structure of wind turbines is mainly affected by low order frequency. Therefore, smartphones with low cost, non-contact, and remote shooting, combined with vision technology, can be used to replace the traditional contact sensors to monitor the dynamic characteristics of wind turbine structures.

*4.3. Shaking Table Test of Wind Turbine Structure*

To verify the monitoring effect of smartphones combined with visual technology on the dynamic characteristics of wind turbine structures, the test used accelerometers,

camera, and smartphone to conduct vibration monitoring by inputting seismic waves into the vibration table to excite the wind turbine structures. The visual measuring points were divided into 10 monitoring points P0~P9 from top to bottom by the wind turbine structure (P0~P4 is on the blade and P5~P9 is on the tower), corresponding to P0, P2, P4, P7, and P9, which are verified with acceleration sensors. Figures 24 and 25 show the response data monitored by the acceleration sensor and smartphone in the flap-wise direction. Since LDS is only used for verification at P0 in the test, to make the article more concise, the full data part of displacement time history shown in Figure 25 was monitored by smartphones. Since the error of smartphones and LDS in time domain can be within 0.1mm, smartphones can be used to monitor the displacement time history of wind turbine structures in the whole field. Table 4 shows the peak values of each measuring point in the two monitoring methods.

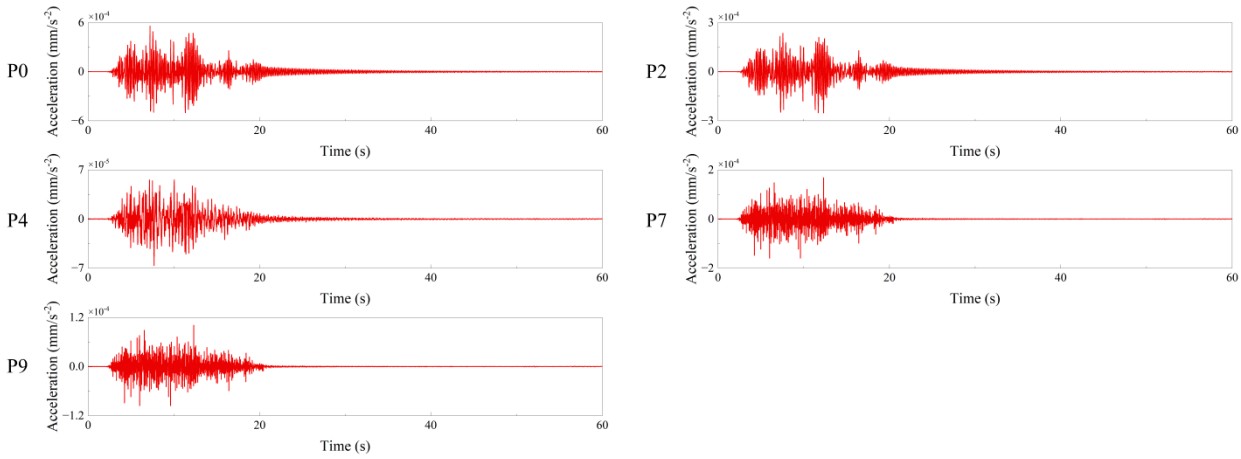

**Figure 24.** Accelerometers monitoring wind turbine structural responses.

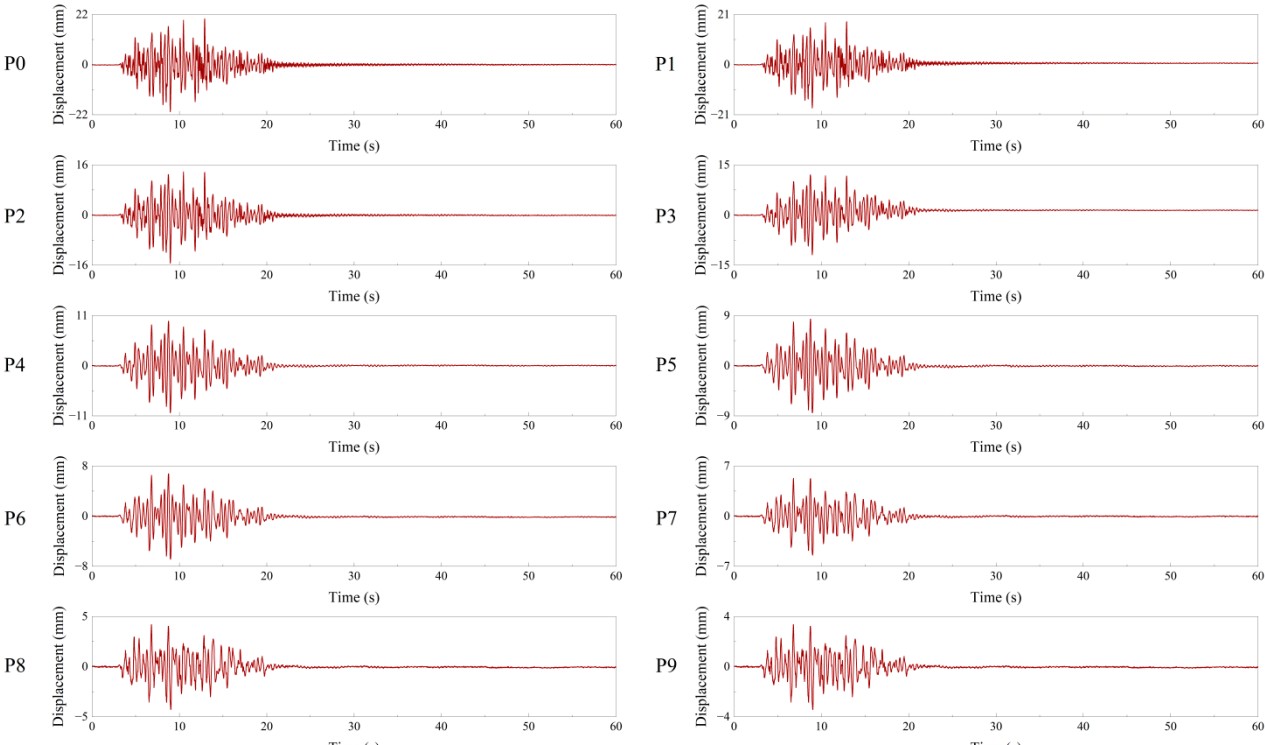

**Figure 25.** Smartphone monitoring wind turbine structure responses.

**Table 4.** Response peak value obtained by two monitoring methods.

| Measuring Points | Accelerometer (mm/s$^2$) | Smartphone (mm) | Measuring Points | Accelerometers (mm/s$^2$) | Smartphone (mm) |
|---|---|---|---|---|---|
| P0 | $5.6341 \times 10^{-4}$ | 20.6585 | P5 | | 8.3780 |
| P1 | | 18.1829 | P6 | | 6.8293 |
| P2 | $2.3902 \times 10^{-4}$ | 15.0244 | P7 | $1.6911 \times 10^{-4}$ | 5.4341 |
| P3 | | 12.0122 | P8 | | 4.2276 |
| P4 | $6.5732 \times 10^{-5}$ | 10.1504 | P9 | $1.0098 \times 10^{-4}$ | 3.4472 |

From the response monitored by the accelerometers shown in Figure 24, the overall structural responses are in accordance with the law of "from big to small". According to the data obtained from P4 measuring point in Table 4, the response of the acceleration sensor at P4 is relatively small. The reason for this is that P4 is located at the top of the tower, and the blade-waving amplitude is relatively large. Therefore, it is reasonable to state that P4 had the smallest acceleration response. Figure 25 shows that the displacement time history monitored by smartphone is relatively stable. It can be seen from Table 4 that smartphones can be used in combination with visual technology for multi-point monitoring. The displacement time histories of each monitoring point are obtained through smartphones, and then the structural modal shapes of wind turbines were calculated by a stochastic subspace identification (SSI) method. The position of the sensor is important when calculating the modal shape [41]. In this paper, five accelerometers and visual sensors are compared, and the final modal shape of the wind turbine structure is shown in Figure 26.

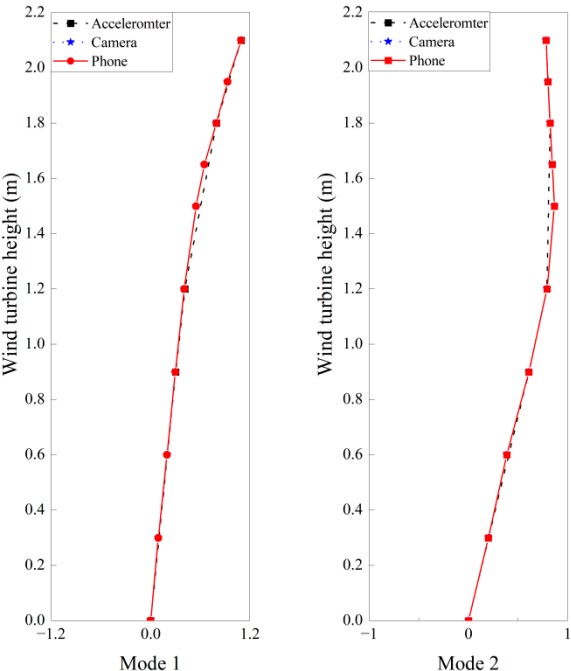

**Figure 26.** Structural modal shape of wind turbine.

It can be seen from Figure 26 that the vibration modes calculated by vision and accelerometers are basically the same. Since there are many measuring points monitored by smartphones combined with vision technology, the vibration modes obtained are smoother. However, the frame rate of the smartphone is 60 fps, and the natural frequency below 30 fps can be monitored at most. Therefore, only the first two modes are obtained. If higher order modal shapes are desired, lenses with higher frame rates should be considered. As the vibration of wind turbine structure in the flap-wise direction only needs the first two steps to be sufficient, the dynamic characteristics of wind turbine structure can be monitored through smartphones combined with visual technology.

## 5. Conclusions

In this study, a target-free dynamic characteristic monitoring method for wind turbine structures using a portable smartphone and optical flow method was proposed. Firstly, the characteristics of smartphones in the monitoring environment were studied to verify the robustness of the proposed algorithm. After that, smartphones in different shooting states were used to monitor the displacement, and high pass filtering combined with adaptive scaling factor was used to process the displacement drift of common smartphone shooting states. Then, the displacement monitoring of smartphone assembling telephoto lens was studied. Finally, the following conclusions can be drawn from the wind turbine structure test verification and result analysis:

(1) The proposed method based on optical flow method for monitoring the target-free dynamic characteristics of wind turbine structures can better identify targets by simulating simple and complex background projects. In addition, the use of smartphones combined with visual algorithms can simultaneously monitor the spatial displacement of the entire blade through ROI clipping.

(2) The method of high pass filtering combined with adaptive scaling factor was adopted to effectively eliminate the displacement drift caused by the two shooting states of standing and slightly walking. The error analysis shows that the final error is less than 2 mm, which can meet the requirements of structural dynamic characteristics monitoring. The smartphone is equipped with a telephoto lens to monitor the displacement of the structure, which effectively expands the method of smartphone to monitor the dynamic characteristics of the structure.

(3) The proposed method for monitoring the dynamic characteristics of wind turbine structures performs well in cooperation with smartphones. Combined with the shaking table test, the results show that using smartphones to monitor the dynamic characteristics of fan structures has higher accuracy in time and frequency domains.

**Author Contributions:** Conceptualization, W.Z. and W.L.; methodology, W.Z. and W.L.; software, W.Z. and B.F.; validation, W.Z., W.L. and B.F.; formal analysis, W.Z.; investigation, W.L.; resources, W.L. and Y.D.; data curation, W.Z.; writing—original draft preparation, W.Z.; writing—review and editing, Y.D.; visualization, W.Z.; supervision, W.L.; project administration, W.L.; funding acquisition, W.L. All authors have read and agreed to the published version of the manuscript.

**Funding:** This research was jointly funded by the National Natural Science Foundation of China (Nos. 52068049, 51908266), the Science Fund for Distinguished Young Scholars of Gansu Province (No. 21JR7RA267), and Hongliu Outstanding Young Talents Program of Lanzhou University of Technology.

**Data Availability Statement:** Some or all data that support the findings of this study are available from the corresponding author upon reasonable request.

**Conflicts of Interest:** The authors declare that they have no known competing financial interests or personal relationships that could have appeared to influence the work reported in this paper.

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
