# Peer review of "Dynamic Characteristic Monitoring of Wind Turbine Structure Using Smartphone and Optical Flow Method"

_buildings, doi:10.3390/buildings12112021_

Round 1

Reviewer 1 Report

The paper presents an investigation of vibration monitoring of a mini wind turbine structure, using a smartphone camera. The research provides a good theoretical background of the Optical Flow Method and conducts several tests on video shooting conditions and tools, which is very interesting.

Major comments:

1. The displacement results obtained by the smartphone appear to be very large, up to around 20 mm in P0 and P1. I wonder if the authors did measure the displacement in another way. what is the error of the predicted displacement?

2. Figure 23 is a very important result of this research, the authors should expand the discussion on it, especially on the apparent stability of the smartphone prediction. 

3. Figure 26 is also very important. The authors should extend the discussion on the method of calculating the vibrational modes through video.

4. All the images should be replaced with high-quality ones.

5. The authors should mention the importance of sensor placement, and research about it such as "Experimental sensitivity analysis of sensor placement based on virtual springs and damage quantification in CFRP composite"

6. Damage detection based on vibrational analysis such as "Residual Force Method for damage identification in a laminated composite plate with different boundary conditions" and "Damage identification in steel plate using FRF and inverse analysis" should be also mentioned.

Minor revisions:

1. Figure 26, right subfigure is mode 12?

2. improve the text expressions that are inexpressive,

such as  "The traditional wind turbine detection is mainly manual" detection? maybe monitoring?

"This paper focuses on the dynamic characteristics monitoring"  maybe "This  paper focuses on monitoring the dynamic characteristics"

3. Divide the long sentences that are hard to read, such as "To determine the relationship between the three-dimensional 174 geometric position of a point on the surface of a space object and its corresponding point 175 in the image, the camera imaging geometric model must be established"

4. Rung the paper through grammar software to fix the small minor mistakes

Author Response

请参阅附件。

Reviewer 2 Report

The dynamic characteristics of existing wind turbine structures are usually monitored using contact sensors, which is not only expensive but also time-consuming and laborious to install. In recent years, with the rapid development of computer vision, monitoring methods based on cameras and UAVs (unmanned aerial vehicles) have been widely used. The topic of this paper is very interesting and falls into the heated topics of SHM nowadays. The overall quality is very good as there are excellent experimental studies included in the manuscript to verify the robustness of the method. The paper can be accepted for publication given the following issues are properly addressed:

(1) The way of determing ROI should be illustraed in a more details;

(2) In Fig. 23, there is no need to use logrithm scale for x (F, Hz) as it makes the PSD look very  strange;

(3) What is the advantages of the optic flow choosen in this study should be further illustrated in the theoretical part;

(4) A more detailed flowchart on the procedure of displacement measurements from the images including carema calibration, ROI selection and optic flow computation, etc. should be included for the better readship of this part. The authors can refer to the paper "Bayesian inference for the dynamic properties of long-span bridges under vortex-induced vibration with Scanlan's model and dense optical flow scheme" for reference.

(5) Please go through the paper carefully as there are still some grammar mistakes in the manuscript.

Round 2

Reviewer 2 Report

All comments raised by the reviewer has been well addressed and the paper can be accepted for publication as is.